# Effective combinatorial immunotherapy for penile squamous cell carcinoma

Tianhe Huang[1,2,3], Xi Cheng[1,2,4], Jad Chahoud[5], Ahmed Sarhan[6], Pheroze Tamboli[7], Priya Rao[7], Ming Guo[8], Ganiraju Manyam [9], Li Zhang[10], Yu Xiang[11], Leng Han [11], Xiaoying Shang[12], Pingna Deng[12], Yanting Luo[1], Xuemin Lu[1], Shan Feng[13], Magaly Martinez Ferrer[14,15], Y. Alan Wang[12], Ronald A. DePinho[12], Curtis A. Pettaway[6] & Xin Lu [1,2 ✉]

Penile squamous cell carcinoma (PSCC) accounts for over 95% of penile malignancies and causes significant mortality and morbidity in developing countries. Molecular mechanisms and therapies of PSCC are understudied, owing to scarcity of laboratory models. Herein, we describe a genetically engineered mouse model of PSCC, by co-deletion of *Smad4* and *Apc* in the androgen-responsive epithelium of the penis. Mouse PSCC fosters an immunosuppressive microenvironment with myeloid-derived suppressor cells (MDSCs) as a dominant population. Preclinical trials in the model demonstrate synergistic efficacy of immune checkpoint blockade with the MDSC-diminishing drugs cabozantinib or celecoxib. A critical clinical problem of PSCC is chemoresistance to cisplatin, which is induced by *Pten* deficiency on the backdrop of *Smad4/Apc* co-deletion. Drug screen studies informed by targeted proteomics identify a few potential therapeutic strategies for PSCC. Our studies have established what we believe to be essential resources for studying PSCC biology and developing therapeutic strategies.

[1] Department of Biological Sciences, Boler-Parseghian Center for Rare and Neglected Diseases, Harper Cancer Research Institute, University of Notre Dame, Notre Dame, IN 46556, USA. [2] Tumor Microenvironment and Metastasis Program, Indiana University Melvin and Bren Simon Cancer Center, Indianapolis, IN 46202, USA. [3] Department of Oncology, The First Affiliated Hospital of Xi'an Jiaotong University, Xi'an, Shaanxi 710061, China. [4] Department of General Surgery, , Ruijin Hospital, Shanghai Jiao Tong University School of Medicine, 200025 Shanghai, China. [5] Department of Cancer Medicine, The University of Texas MD Anderson Cancer Center, Houston, TX 77030, USA. [6] Department of Urology, The University of Texas MD Anderson Cancer Center, Houston, TX 77030, USA. [7] Department of Pathology, The University of Texas MD Anderson Cancer Center, Houston, TX 77030, USA. [8] Department of Pathology/Lab Medicine, The University of Texas MD Anderson Cancer Center, Houston, TX 77030, USA. [9] Department of Bioinformatics & Computational Biology, The University of Texas MD Anderson Cancer Center, Houston, TX 77030, USA. [10] Department of Environmental Health, University of Cincinnati, Cincinnati, OH 45267, USA. [11] Department of Biochemistry and Molecular Biology, The University of Texas Health Science Center at Houston McGovern Medical School, Houston, TX 77030, USA. [12] Department of Cancer Biology, The University of Texas MD Anderson Cancer Center, Houston, TX 77030, USA. [13] Mass Spectrometry Core Facility, School of Life Sciences, Westlake University, Hangzhou, 310024 Zhejiang, China. [14] Department of Pharmaceutical Sciences, School of Pharmacy, University of Puerto Rico, San Juan, PR 00936, USA. [15] University of Puerto Rico Comprehensive Cancer Center, Medical Sciences Campus, San Juan, PR 00936, USA. ✉email: xlu@nd.edu

Penile cancer can constitute up to 10% of male malignancies in some African, Asian and South American areas[1]. While it represents ~0.5% of all cancers among men in the United States and other developed countries, penile cancer often results in devastating disfigurement and only half of the patients survive beyond 5 years[1]. PSCC accounts for over 95% of penile malignancies[1]. Risk factors associated with PSCC incidence include the absence of neonatal circumcision, phimosis, chronic inflammation, poor hygiene, tobacco use, and infection with human papillomavirus (HPV)[2]. A recent large meta-analysis reports a pooled HPV prevalence of 50.8% in penile cancer[3]. HPV infection is more common in carcinomas in situ than in invasive forms[4], and the presence of high-risk HPV confers a survival advantage in patients with PSCC[5]. It is possible that HPV-positive and HPV-negative PSCC develop through certain convergent mechanisms, which may be leveraged for therapies. Expression alterations for some genes have been documented in PSCC, such as RAS, MYC, EGFR, E-cadherin, COX2, IGF1R, and MMP2/9[6]. However, the precise molecular mechanisms governing PSCC initiation and progression remains elusive, mainly due to the shortage of experimental tools. Recently, a number of PSCC cell lines have been developed[7–9]. However, the generation of genetically engineered mouse (GEM) models of PSCC has not been reported.

Standard treatment regimens for penile cancer are limited to surgery, radiation, and chemotherapy[1,10,11]. The mainstay of systemic therapy for advanced PSCC is cisplatin-based chemotherapy, with average response rates between 15 to 55% and median overall survival (OS) ranging between 5 and 12 months[1,10–12]. Patients whose PSCC progresses or recurs after front-line cisplatin-based chemotherapy experience poor responses to salvage treatments (OS < 6 months)[13]. Therefore, there is an urgent need to identify molecular mechanisms for chemoresistance of PSCC and seek other approaches for metastatic PSCC. Immunotherapy using immune checkpoint blockade (ICB), such as anti-CTLA4 and anti-PD1 antibodies, has revolutionized cancer treatment and has generated durable therapeutic responses in a significant subset of patients across a variety of cancer types. Currently clinical trials are recruiting patients to assess the role of ICB agents in penile cancer specifically or a rare tumor cohort that includes penile cancer. Examples include pembrolizumab for advanced PSCC (NCT02837042) and ipilimumab plus nivolumab for advanced rare tumors including penile cancer (NCT03333616, NCT02834013). Developing resources for PSCC such as GEM models will be critical to evaluate the efficacy of ICB in a meaningful preclinical setting and, perhaps more importantly, to predict potential resistance mechanisms for ICB and design corresponding combination therapy strategies.

Here, we report the generation and characterization of two GEM models of PSCC through the manipulation of signaling pathways relevant to HPV carcinogenesis mechanisms and clinical evidence of PSCC. The murine PSCC displays strong immune gene signatures and infiltration, consistent with clinical evidence gathered by bioinformatics analysis and immunohistochemistry validation. Intratumoral immunosuppressive myeloid cell infiltration suggests the benefit of combining targeted therapy and immunotherapy to achieve maximal clinical efficacy. This illuminates a testable clinical trial hypothesis for combination therapy in the treatment of lethal PSCC.

## Results

### Establishment of genetically engineered mouse model of PSCC.
Androgen receptor (AR) is expressed in the epithelium of the developing human fetal penis[14]. We stained human primary penile tumor samples (n = 8, Supplementary Table 1) and observed pronounced nuclear AR expression in adjacent normal epithelium (5/5 cases) but not in the malignant area (8/8 cases, Fig. 1a). There is also persistent AR expression in the rodent penis[15], which we further confirmed in the penis of wild type adult male mice (Fig. 1b). This result prompted us to hypothesize that a GEM model employing an AR-responsive Cre driver in conjunction with LoxP alleles of tumor suppressor genes with relevance to human penile cancer could form penile cancer in mice. The Cre-expressing line PB-Cre4 carries the Cre gene under the control of a strong AR-responsive promoter and is frequently used to manipulate gene expression in the mouse prostate[16]. To determine if PB-Cre4 drives recombination of mouse penile epithelium, we crossed PB-Cre4 to the fluorescence reporter allele mTmG[17] and the penis clearly showed GFP expression in the epithelium (Fig. 1c and Supplementary Fig. 1a). AR$^+$ nuclei overlapped with GFP$^+$ epithelial cells (Supplementary Fig. 1b). To examine genetic changes that may cause penile tumorigenesis, we analyzed several previously developed prostate tumor models that used PB-Cre4, including Pten[18], Pten Smad4[19], and Pten p53 Smad4[20]. Normal histology of the penile epithelium was observed for these mice even at the age when they succumbed to the prostate tumor burden (Supplementary Fig. 1c), indicating loss of function for Pten, p53 or Smad4 (individually or in combination) is insufficient to drive penile tumorigenesis.

Next, we focused on two pathways, Wnt/Apc/β-catenin and TGFβ/Smad pathways, based on their relevance to oncogenic HPV. Oncogenic HPV encodes two oncoproteins E6 and E7, which bind and facilitate degradation of p53 and Rb, respectively[21]. While E6 and E7 can immortalize cells in vitro, they are not sufficient for tumor development in vivo[21], suggesting that additional mechanisms are necessary for tumor initiation. E6 binds to cellular partner E6AP to stabilize β-catenin and stimulate Wnt signaling[22], and knockdown of E6 and E7 expression induces a substantial reduction of nuclear β-catenin and TCF transcriptional activity[23]. E7 blocks TGF-β induced transcription and growth inhibition by directly binding to Smad3 and Smad4 thus interfering with their interaction[24]. E6 has a similar activity[25]. To recapitulate these signaling effects from E6/E7, we reasoned that Wnt/β-catenin activation and TGFβ/Smad pathway inactivation in the mouse penile epithelium might lead to penile tumorigenesis. We used conditional null alleles of Apc and Smad4 to test our hypothesis. PB-Cre4$^+$ Smad4$^{L/L}$ mice had no prostate lesion[20], nor did we observe them to show any abnormality of the penis (Supplementary Fig. 1d). PB-Cre4$^+$ Apc$^{L/L}$ mice developed prostate hyperplasia with squamous metaplasia as early as 7 months of age[26], and the penis of these mice was normal at 3.5 months of age but displayed mild dysplasia with hyperkeratosis at 12.5 months of age (Fig. 1d and Supplementary Fig. 1d). The epidermal layer was moderately multiplied, yet no SCC pathology was found (Supplementary Fig. 1d). Penile prolapse (paraphimosis) is a condition in which the penis no longer retracts back into the prepuce, and can be used as a sign to indicate penile tumor formation in mice. In total, 0/19 PB-Cre4$^+$ Smad4$^{L/L}$ mice and 1/18 PB-Cre4$^+$ Apc$^{L/L}$ mice exhibited penile prolapse (Fig. 1e). The only prolapsed penis of a PB-Cre4$^+$ Apc$^{L/L}$ mouse was caused by excessive keratinization. Remarkably, when Smad4 and Apc were co-deleted, PB-Cre4$^+$ Smad4$^{L/L}$ Apc$^{L/L}$ mice (SA genotype) developed penile prolapse at 100% penetrance (Fig. 1e). When crossed to mTmG allele (SAm genotype), GFP$^+$ solid tumor nodules formed in mouse penis (Fig. 1f). At the histological level, the tumors contain conspicuous keratin pearls, which are pathognomonic for SCC (Fig. 1g). We noted that, by the time the penile tumors formed in these mice (median 17.2 weeks), the prostate remained normal (Fig. 1h). Given the AR expression by normal penile epithelium (Fig. 1b), we determined whether AR signaling was required to sustain SA

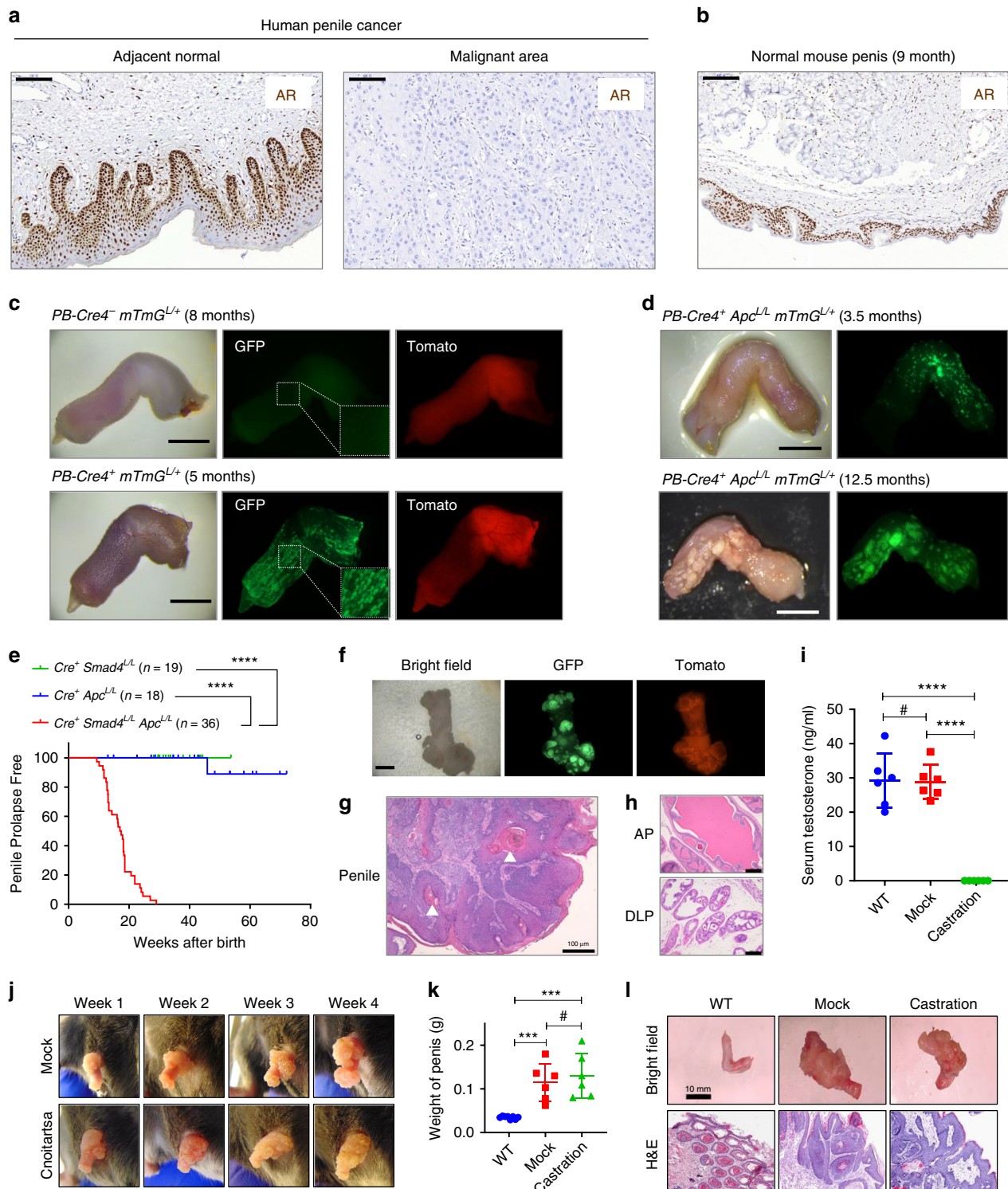

**Fig. 1 Smad4 and Apc co-deletion leads to penile squamous cell carcinoma in mice. a**, **b** IHC stain for AR in human penile tumors ($n = 8$) and normal mouse penis ($n = 5$). Scale bar 100 μm. **c**, **d** Morphology and fluorescence for resected mouse penises of indicated genotype and age. Scale bar 2 mm. **e** Penile prolapse free survival curves for mice of three genotypes with $n$ indicated. ****$P < 0.0001$, log-rank test. **f** Morphology and fluorescence for resected mouse penis of SAm genotype at 6.4 months of age, with GFP⁺ tumor nodules clearly visible. Scale bar 2 mm. **g**, **h** H&E stain of penis and prostate lobes (anterior AP, dorsolateral DLP) of SA mouse at 6.8 month of age. White arrows denote keratin pearls. Scale bar 100 μm. **i** SA males at 8–12 weeks underwent castration or mock surgical procedure. After 4 weeks, the mice were euthanized. The serum testosterone was measured with ELISA for the mice and age-matched wild type (WT) males were used as control ($n = 6$ for each group). **j** Representative images of the mouse penis at each week post-surgery. **k**, **l** Weight, representative image and H&E staining of resected penises from WT control or SA mice 4 weeks post-surgery. Scale bar 10 mm and 200 μm ($n = 8$ for WT, $n = 6$ for mock and castration). In **i**, **k**, data represent mean ± SD. #$P > 0.05$, ***$P < 0.001$, ****$P < 0.0001$, two-sided Student's $t$ test.

penile tumor growth by performing mock procedure or surgical castration on 8–12-week-old SA males and following the penile tumor progression for 4 weeks. Castration depleted serum testosterone level (Fig. 1i), yet elicited undetectable effect on penile tumor growth (Fig. 1j–k), histology (Fig. 1l), proliferation marker Ki67, or apoptosis marker cleaved caspase-3 (Supplementary Fig. 1e). We conclude that AR signaling is dispensable for sustaining PSCC. In summary, the first GEM model of PSCC was developed by co-deleting *Apc* and *Smad4* in the penile epithelium.

**Transcriptomic analysis of mouse penile cancer**. To identify differentially expressed genes between normal and cancerous mouse penis, we performed RNA-seq on penile tumors from 5-month-old SA mice and penile epithelium from age-matched wild type (WT) mice. Significant genes were defined by using a cut-off of 0.01 on the Benjamini-Hochberg corrected *P*-value and an absolute log$_2$(fold change) value of at least 2 (i.e. fold change ≥ 4). We detected a total of 1660 significantly upregulated genes and 642 downregulated genes in SA compared with the WT penis (Fig. 2a, Supplementary Fig. 2a, and Supplementary Table 2). To identify significantly enriched pathways by SA tumors, Ingenuity Pathway Analysis (IPA) was used to analyze the differentially expressed genes and identified 102 Ingenuity Canonical Pathways with *P*-value < 0.05 (Supplementary Table 3). Of note, the top ranked two pathways are Granulocyte Adhesion and Diapedesis and Agranulocyte Adhesion and Diapedesis, because of upregulation of many cytokines and cytokine receptors suggestive of an inflammatory phenotype (Supplementary Table 4). Matrix Metalloproteases and Eicosanoid Signaling were significantly enriched (Supplementary Tables 3 and 4), underscoring the invasive SCC histology and supporting the inflammation of SA tumors. Both Wnt/β-catenin Signaling and FGF Signaling were hyperactivated in SA tumors as a result of genetic loss of *Apc* and dramatically enhanced expression of FGF ligands, respectively (Supplementary Tables 3 and 4). IPA can provide analyses to identify putative upstream regulators to account for gene expression changes, which listed β-catenin as the top putative upstream regulator (Supplementary Table 5). Massive transcriptional regulation by β-catenin complex (Fig. 2b) was expected because of *Apc* loss, with confirmed nuclear concentration in SA tumors compared with WT penis (Fig. 2c). Enhanced β-catenin localization in the nucleus and cytoplasma was also validated in about one third of human PSCC samples using tissue array analysis ($n = 191$ total, Supplementary Figs. 2b, c). Two of β-catenin transcriptional targets, *Sox2* and *Ptgs2*, were upregulated by 137-fold and 9-fold respectively at the RNA level. Sox2 was shown to be a master regulator for cancer stem cells in skin SCC[27]. Ptgs2, better known as Cox2, converts arachidonic acid to prostaglandins, which in turn induce inflammatory reactions. Significant upregulation of Sox2 and Cox2 in SA tumors at the protein level was confirmed by immunohistochemistry (IHC) (Figs. 2d, e). By a separate algorithm in IPA to estimate master regulators to explain transcriptomic changes and pathway enrichment, Cox2 was identified as the top regulator to account for cytokine upregulation and the phenotype Adhesion of immune cells (Fig. 2f and Supplementary Table 6). COX2 upregulation is highly relevant to PSCC as previously shown[28]. Western blot further confirmed upregulation of Cox2 and Sox2 in SA tumors relative to penile tissues from WT and *PB-Cre4+ Smad4$^{L/L}$* mice (Fig. 2g). At the protein level, we also detected upregulated cyclin D1 and phospho-Rb (Ser780) signals in SA tumors (Fig. 2g). Intensified cyclin D1 expression is expected given its known regulation by Smad4[20] and β-catenin/LEF1 complex[29]. Cyclin D1 overexpression and correlated Ki67

expression was reported in PSCC[30] and confirmed in our model (Fig. 2h). Using a penile cancer cell line, SA1, established from an SA penile tumor, we generated shRNA-mediated *Sox2* knockdown (Fig. 2i), which attenuated subcutaneous tumor growth of SA1 (Fig. 2j–l). Together, transcriptomic profiling revealed extensive signaling changes associated with PSCC formation in mice and, in particular, signaling molecules involved in inflammation and immunity.

**Infiltration of immunosuppressive myeloid cells**. Transcriptional profiling suggests massive immune response in the SA penile tumors. To examine the immune infiltration, we dissociated penile tumors from 4–5 month-old SA mice and normal penile epithelium from Cre- littermates, and catalogued the intratumoral immunocytes by mass cytometry (CyTOF) using a 35-antibody panel[31]. Consistent with the enrichment of various immune-related signaling pathways, there was a 7-fold increase of CD45+ immune cells (as fraction of all live cells) in mouse PSCC compared with normal penis (Fig. 3a). When quantified as percentage of all immune cells, there were significant decreases of CD8+ T cells, NK cells, B cells and tumor-associated macrophages (TAMs), yet there was a dramatic increase from ~1% to ~50% of CD11b+ Gr1+ myeloid cells (Fig. 3b, c, and Supplementary Fig. 3). We isolated the CD11b+ Gr1+ cells from established SA tumors and demonstrated their strong ability to suppress T cell proliferation (Fig. 3d). This result indicates that the intratumoral CD11b+ Gr1+ cells belong to myeloid-derived suppressor cells (MDSCs) that have been recognized as a major immunosuppressive population to induce T cell tolerance in solid tumors. Recent work from others and us indicated that PI3K signaling is crucial to mediate the immunosuppressive activity of myeloid cells[31–33]. In accordance with this notion, viSNE analysis of the CyTOF data showed stronger phospho-PI3K, phospho-mTOR and phospho-S6 signals in the CD11b+ Gr1+ population compared with other immune cells (Fig. 3c). This result suggests that targeting PI3K signaling may be an effective approach to attenuate MDSC activities in the mouse PSCC and restore the response to immune checkpoint blockade (ICB) therapy. Given that the infiltrating T cells in the SA tumors are positive for PD1 (Fig. 3e), it is rational to test whether co-targeting MDSCs synergizes with the ICB therapy.

**Combining targeted therapy and immunotherapy**. To examine the efficacy of ICB in the SA model, we randomized SA males with established PSCC at 4–5 month old to receive isotype IgG (control) or an anti-PD1/anti-CTLA4 antibody cocktail (ICB) at doses described recently[31]. The treatment lasted for one month and affected minimally on tumor weight (Fig. 4a, b). To enhance ICB, we focused on two FDA-approved drugs, cabozantinib and celecoxib. Cabozantinib is a multi-targeted tyrosine kinase inhibitor and recently shown to block PI3K signaling in MDSCs to enable efficacy from ICB in metastatic prostate cancer[31]. Celecoxib is a selective COX2 inhibitor and highly relevant to our study, because Cox2 was upregulated in the SA tumors (Fig. 2e) and identified as the top putative master regulator for inflammation-related phenotypes (Fig. 2f). Cyclooxygenase-dependent tumor immune evasion is through myeloid cell reprograming and Cox2 blockade with celecoxib synergizes with anti-PD1 or dendritic cell immunotherapy in syngeneic models of a few cancer types[34,35].

While the targeted therapy drugs, similar to ICB alone, caused marginal effect on penile tumor progression and endpoint weight, the combinations (cabozantinib plus ICB, celecoxib plus ICB) led to eradiation of most tumor nodules (Fig. 4a, b). At the cellular level, cabozantinib or celecoxib was sufficient to significantly reduce the infiltration of myeloid cells positive for CD11b (Fig. 4c

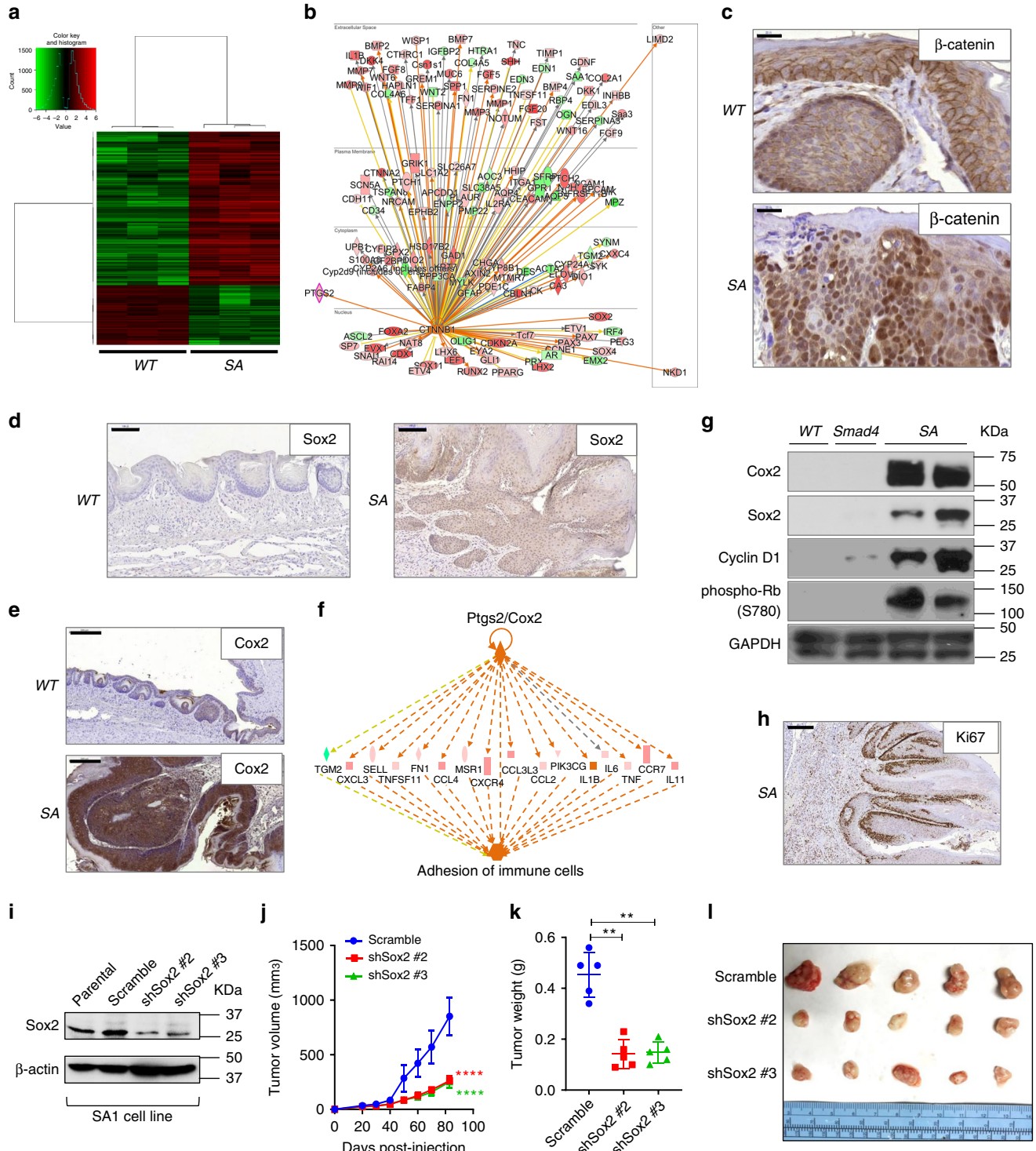

**Fig. 2 Transcriptomic analysis reveals activation of β-catenin signaling and inflammatory pathways in mouse PSCC. a** Hierarchical clustering of differentially expressed genes between WT and SA mouse penile samples ($n = 3$ for each genotype). **b** Gene regulation network by β-catenin (i.e. CTNNB1), the top upstream regulator identified by IPA to account for differential gene expression changes in SA tumors compared with WT penis. Red and green colors indicate upregulation and downregulation, respectively. **c–e** IHC stain for β-catenin, Sox2 and Cox2 in WT and SA penis, respectively. Scale bar 20 μm for **c**, 100 μm for **d**, 200 μm for **e**. **f** Mechanistic network for the top ranked master regulator Ptgs2/Cox2 by IPA to illustrate its effect on cytokine expression regulation and function in immune cells. **g** Western blot showing differential protein expression in penile tissues from WT, *PB-Cre4+ Smad4L/L* (Smad4) and SA mice. **h** IHC stain for Ki67 in SA penile tumor. Scale bar 200 μm. **i** Sox2 expression silenced by two independent shRNA in SA1 cell line, detected by western blot. **j** Growth curves of subcutaneous tumors formed by control or Sox2 knockdown sublines of SA1 in nude mice ($n = 5$). **k, l** Weight and gross images of subcutaneous tumors formed by control or Sox2 knockdown sublines of SA1 at endpoint (Day 83) ($n = 5$). In **j, k**, data represent mean ± SD. **$P < 0.01$, ****$P < 0.0001$, two-sided Student's $t$ test.

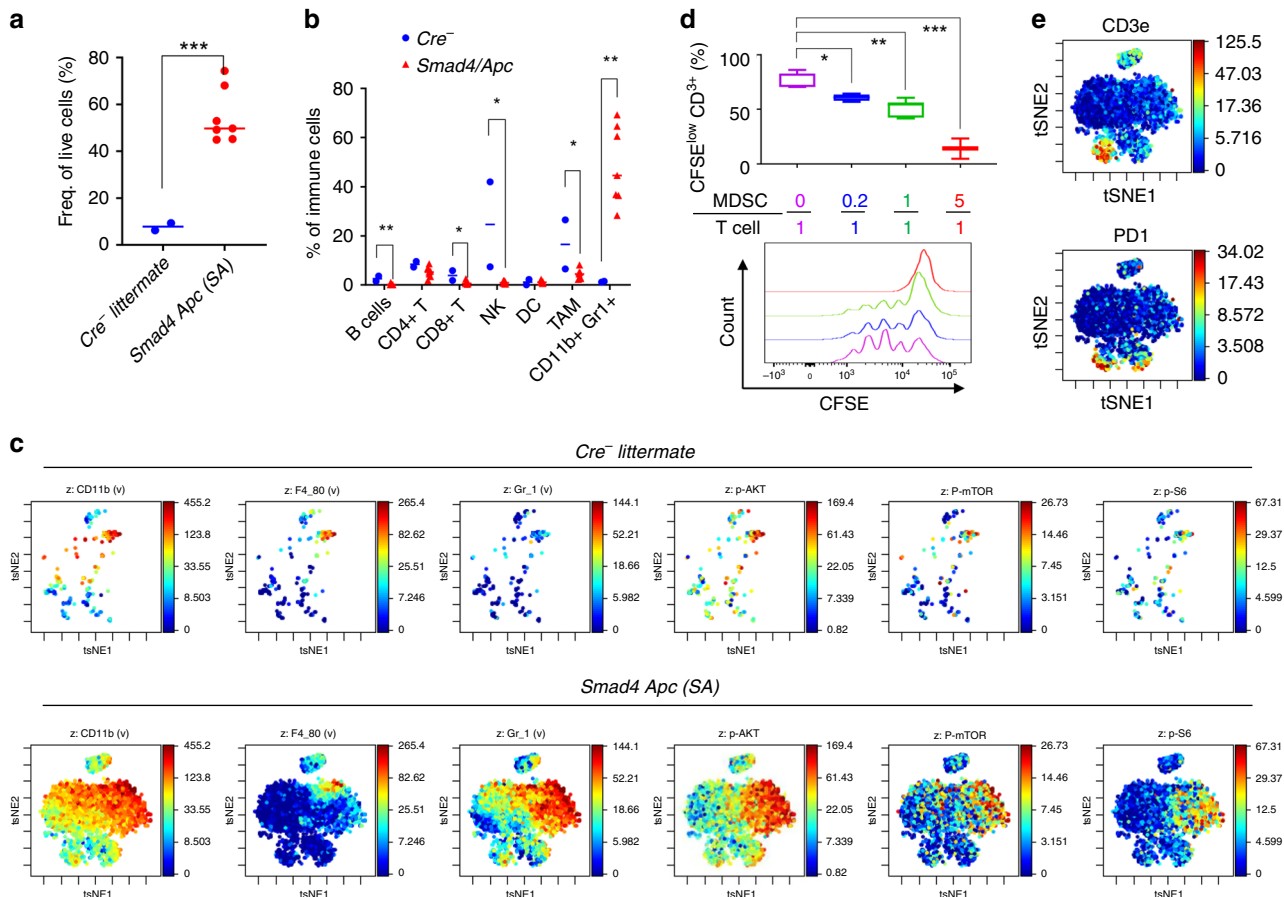

**Fig. 3 Infiltration of CD11b$^+$ Gr1$^+$ immunosuppressive myeloid cells in mouse penile cancer. a** Frequency of CD45$^+$ immune cells of all live cells in penile tissues from Cre$^-$ littermates (control, $n = 2$) and SA mice ($n = 7$, 4–5 months of age), with data from CyTOF. **b** Frequency of indicated immune cell subpopulations of all immune cells for the same sets of penile samples as in **a**. **c** Representative viSNE plots of indicated markers for the same sets of penile samples as in **a**. Note that the sparse dots and numerous dots for Cre$^-$ littermate or SA mouse, respectively, reflect the vast difference of the total intratumoral immune cell frequency of the two genotypes. **d** Activity of CD11b$^+$ Gr1$^+$ cells from SA tumors to suppress proliferation of normal spleen T cells stimulated with anti-CD3/CD28 antibodies. T cells were preloaded with CFSE whose signals declined as T cells divided. Higher MDSC: T cell ratio led to lower percentage of CFSE$^{low}$ T cells ($n = 4, 4, 6$, and 3 for the ratios in order). **e** Representative viSNE plots of CD3e and PD1 (CD279) for SA tumors, showing the partial overlap of the two signals. In **a**, **b**, all data are shown with the line representing the median. In **d**, box plots visualize the five-number summary of a data set (minimum, lower quartile, median, upper quartile and maximum). *$P < 0.05$, **$P < 0.01$, ***$P < 0.001$, two-sided Student's $t$ test.

and Supplementary Fig. 4a) and Ly6G (Fig. 4d, e), consistent with the reported effect from these drugs. On the other hand, ICB potently decreased the number of Foxp3$^+$ regulatory T cells (T$_{regs}$) in the tumors (Fig. 4f and Supplementary Fig. 4b). Cabozantinib plus ICB, or celecoxib plus ICB treatment led to simultaneous reduction of both myeloid cells and T$_{regs}$ (Fig. 4c, d, and f), explaining the synergistic efficacy. Our results demonstrate that the SA model can serve as the platform to discover combination strategies for immunotherapy of spontaneous PSCC.

**Pten deletion confers resistance to cisplatin.** PI3K/Akt/mTOR pathway is relevant to both HPV-positive and HPV-negative PSCC. Infection by HPV 16, the most common high-risk HPV type in PSCC, induces PI3K/Akt/mTOR pathway which plays a critical role mediating the cellular entry of the virus[36]. Clinically, PIK3CA mutation is found to happen in 29% of PSCC cases[37], and PTEN expression loss and PI3K/Akt/mTOR pathway activation is common in PSCC independently of HPV infection[38,39]. These findings prompted us to cross the SA mice with the conditional null allele of *Pten* to generate the *PB-Cre4$^+$ Smad4$^{L/L}$ Apc$^{L/L}$ Pten$^{L/L}$* mice (SAP genotype). Homozygous, but not heterozygous, loss of *Pten* dramatically accelerated the formation of

penile tumors (Fig. 5a). Median prolapse survival was 17.2 weeks and 8.1 weeks for SA and SAP mice, respectively. As expected, *Pten* loss in the penis augmented the phospho-Akt signals, shown in *PB-Cre4$^+$ Pten$^{L/L}$* mice and more pronounced in the SAP mice (Fig. 5b). In contrast to the SA model, the SAP mice co-develop highly aggressive prostate tumors at a similar timeframe as the penile tumors. To address the question whether PSCC forms in SAP mice independently of the prostate tumors, we took two approaches. First, we performed radical prostatectomy on SAP male at 8 weeks of age and followed the penile tumor progression for 3 more weeks (Fig. 5c). Without the prostate, SAP penile tumors continued to grow and at the endpoint stained strongly with Ki67 and weakly with cleaved caspase-3 (Fig. 5d). Second, we established a PSCC cell line SAP1 from the SAP model and orthotopically injected into the penile and prostate tissues individually or simultaneously to evaluate whether orthotopic penile tumor growth was affected by orthotopic prostate tumors. The result showed robust injection-site-dependent orthotopic tumor formation (Fig. 5e) and the orthotopic penile tumors remained salient SCC features such as keratin pearl and positivity for cytokeratin-5 (Fig. 5f). Orthotopic injection of SA1 (the cell line derived from SA tumors) also generated PSCC, albeit with a

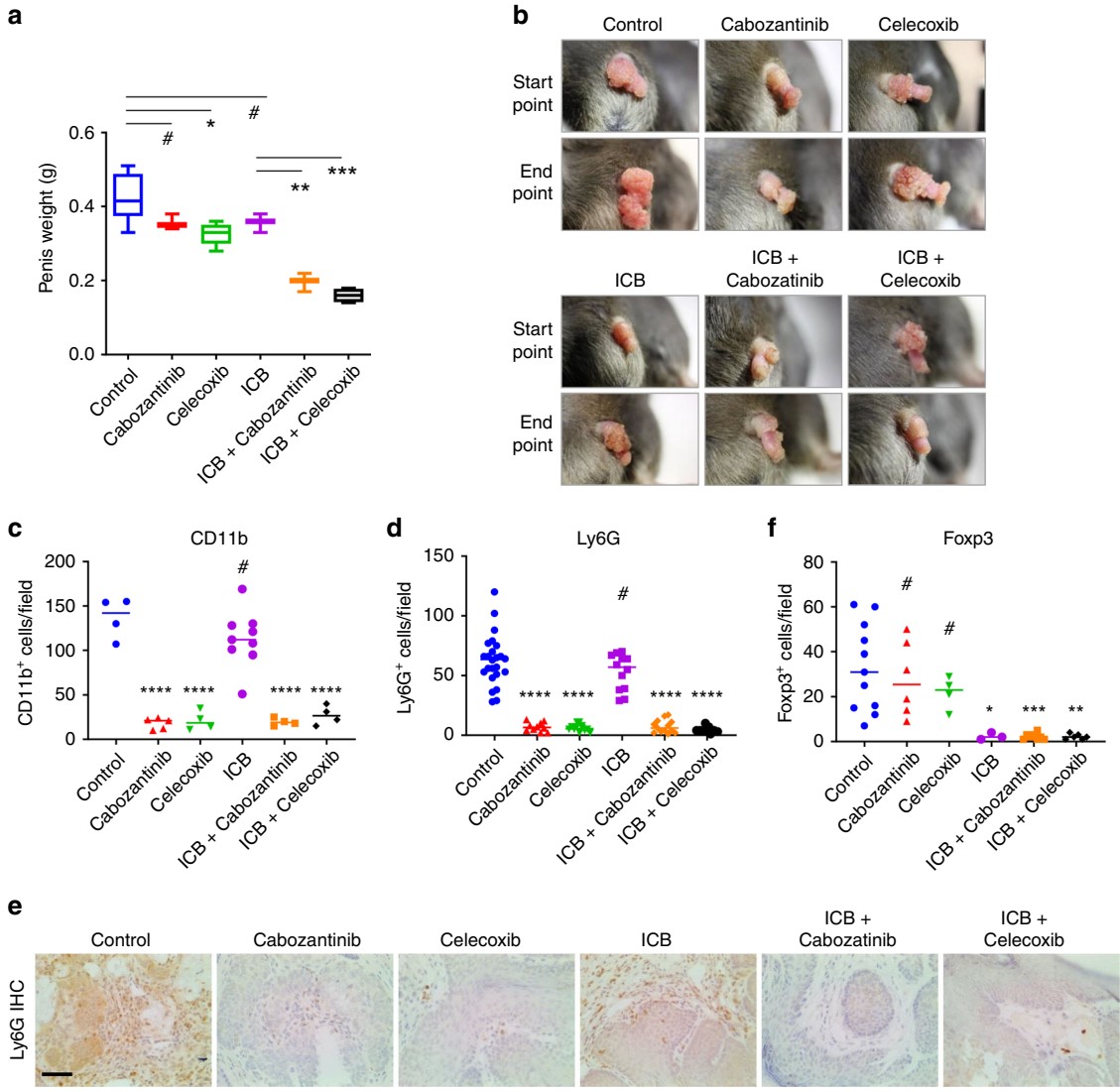

**Fig. 4 Combined targeted therapy and immunotherapy for spontaneous PSCC in mice. a** Weight of resected penis from SA mice at the endpoint of the treatment course with single or combination therapies ($n = 6, 3, 5, 3, 3$, and 4, respectively). **b** Representative images of penises for the SA mice before and after the 1-month treatment course. **c, d** Quantification of IHC staining for CD11b and Ly6G, markers of tumor infiltrating granulocytic MDSCs, using SA tumor samples treated differently (independent IHC images for CD11b: $n = 4, 5, 4, 9, 4$, and 4, respectively; independent IHC images for Ly6G: $n = 24, 10, 12, 12, 13$, and 9, respectively). **e** Representative IHC images of Ly6G from the experiment in **d**. Scale bar 50 μm. **f** Quantification of IHC staining for Foxp3, marker of $T_{regs}$, using SA tumor samples treated differently (independent IHC images: $n = 11, 6, 4, 3, 8$, and 6, respectively). In **a**, box plots visualize the five-number summary of a data set (minimum, lower quartile, median, upper quartile and maximum). In **c, d**, and **f**, all data are shown with the line representing the median. #$P > 0.05$, *$P < 0.05$, **$P < 0.01$, ***$P < 0.001$, ****$P < 0.0001$, two-sided Student's $t$ test.

longer latency (Supplementary Fig. 5a). Therefore, these data rule out that the PSCC formed in the SAP model is due to distant effects or metastasis from the concomitantly formed prostate tumors.

There are four general modes of resistance mechanisms for cisplatin[40], and PI3K/Akt activation stands as an important post-target mechanism[41,42]. To evaluate the effect of *Pten* deletion on cisplatin response in penile cancer, we subjected SA and SAP mice with established penile tumors to cisplatin dosed at 10 mg/kg weekly for a month. Cisplatin generated potent anti-tumor effect for SA mice, but affected SAP tumors minimally (Fig. 5g, h). This anti-tumor effect on SA but not SAP models was likely the result of distinct sensitivity of the two models to cisplatin-induced apoptosis (Supplementary Fig. 5b). Overall, the SAP model provides an essential tool to study cisplatin resistance.

**Targeted proteomics profiling and drug screen.** Reverse-phase protein arrays (RPPA) is a technology for high-throughput protein activity measurement across many signaling pathways involved in cancer[43]. We performed RPPA profiling and observed a genotype-correlated protein expression pattern (Fig. 6a and Supplementary Table 7). Penises from wild type mice clustered together with those from *PB-Cre+ Apc^{L/L}* and *PB-Cre+ Pten^{L/L}* mice (no tumors), whereas SA and SAP tumors clustered in separate branches. Effect from cisplatin treatment (Fig. 5c) was notable by the separation of treated tumor from untreated tumors in both genotypes. Despite the separation of SA and SAP samples in the clustering, there was a large overlap of differentially expressed protein species in SA and SAP tumors compared with WT penises (Fig. 6b and Supplementary Table 8). Most of the overlapped proteins were up- or down-regulated in the same

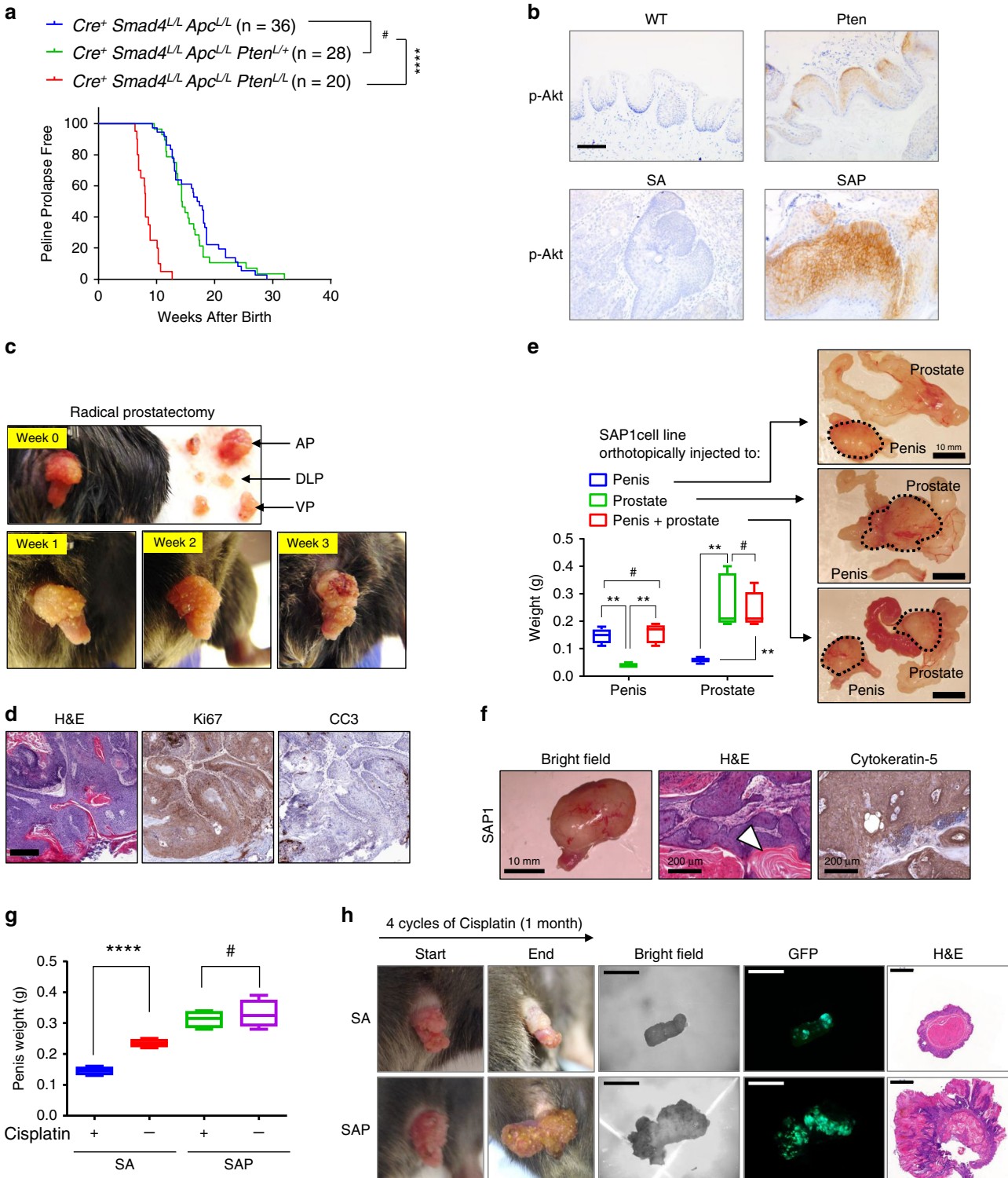

**Fig. 5 SAP mice as a model to study chemoresistance of penile cancer. a** Penile prolapse free survival curves for mice of three genotypes with *n* indicated. ****$P < 0.0001$, #$P > 0.05$, log-rank test. **b** Representative IHC images of phospho-Akt (Ser473). Scale bar 100 µm. **c** SAP mouse at 8 weeks old with all prostate lobes removed (anterior AP, dorsolateral DLP, ventral VP) and penile tumor followed for 3 weeks before sacrifice. **d** Representative H&E and IHC images of Ki67 and cleaved caspase-3 (CC3) of the penile tumor from the prostate-removed SAP mouse (*n* = 3). Scale bar 200 µm. **e** Orthotopic injection of SAP1 cells under the penile epithelium, or into the prostate gland or at both sites of 6-week *Rag1*$^{−/−}$ males (*n* = 5). The mice were euthanized at Day 30 post-injection for organ weight measurement and imaging. Scale bar 10 mm. (**f**) Representative penile tumor formed by penile injection of SAP1 with H&E staining and IHC of cytokeratin-5 (*n* = 5). Scale bar 10 mm and 200 µm. **g** Weight of resected penis from SA and SAP mice at the endpoint of cisplatin treatment course (*n* = 4 for each group). **h** Representative images and H&E staining of penises for the SA and SAP mice before and after the 1-month cisplatin treatment. Scale bars 5 mm (bright field and GFP); 1 mm (H&E). In **e**, **g**, box plots visualize the five-number summary of a data set (minimum, lower quartile, median, upper quartile and maximum). #$P < 0.05$, **$P < 0.01$, ****$P < 0.0001$, two-sided Student's *t* test.

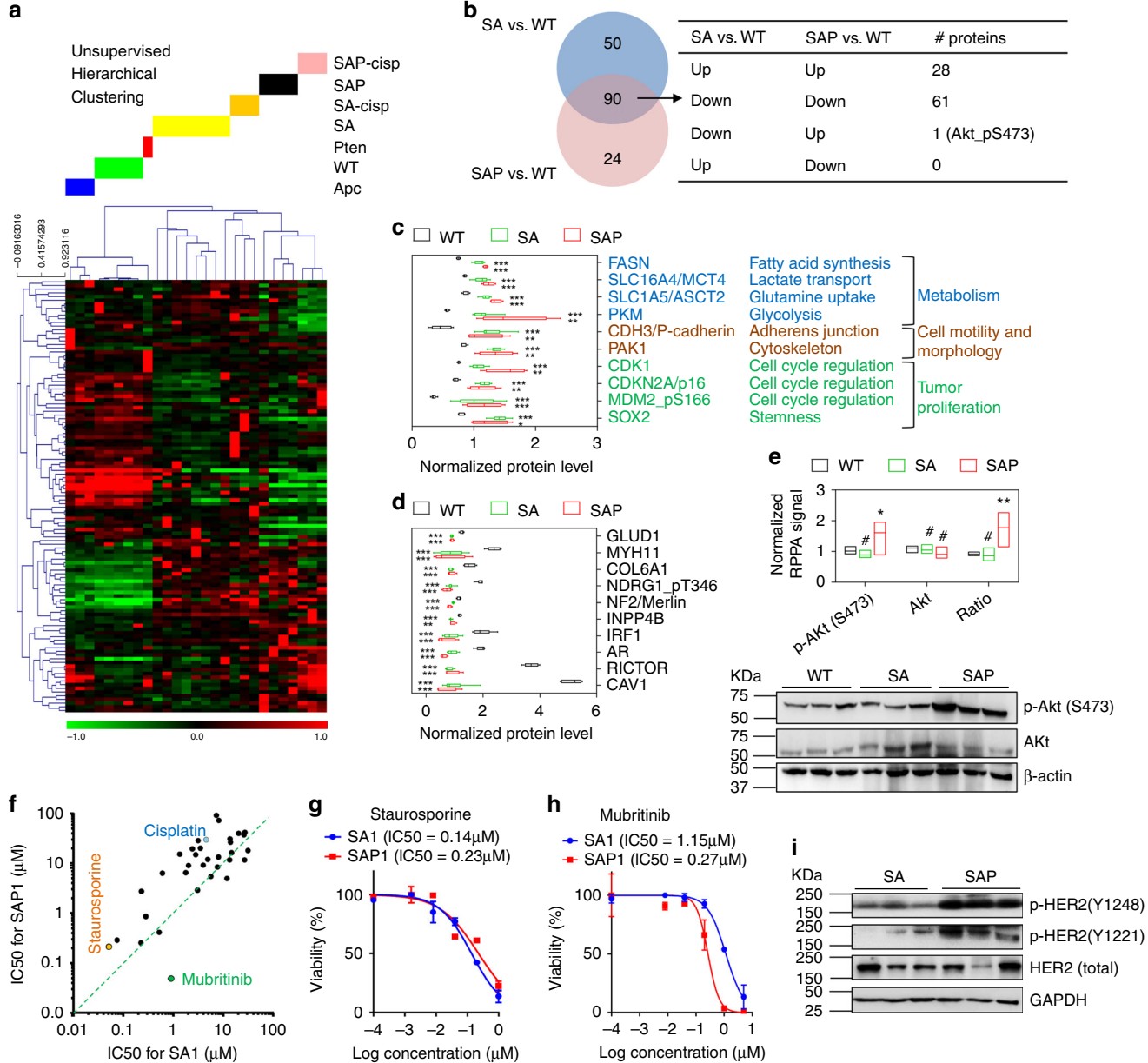

**Fig. 6 RPPA profiling and drug screen using mouse penile cancer cells. a** Unsupervised hierarchical clustering of penile samples from mice of denoted genotypes and treatment (cisplatin) based on normalized RPPA data. One-way ANOVA identified 117 differentially expressed proteins ($P < 0.05$), which were used for the clustering. **b** Venn diagram showing the relationship of differentially expressed proteins between SA vs. WT and SAP vs. WT comparisons, with data from RPPA. two-sided Student's $t$ test was used to select significant proteins for each comparison ($P < 0.05$). **c** Normalized RPPA signals of selected proteins upregulated in both SA and SAP tumors compared with WT penises, with protein names and functions shown ($n = 5$, 8, and 4 for WT, SA and SAP, respectively). **d** Normalized RPPA signals of selected proteins downregulated in both SA and SAP tumors compared with WT penises ($n = 5$, 8, and 4 for WT, SA and SAP, respectively). **e** Normalized RPPA signals and ratios for phospho-Akt (S473) and total Akt in WT, SA and SAP penises ($n = 5$, 8 and 4 for WT, SA and SAP, respectively), validated by western blot of the tissues. **f** Dot plot comparing IC50 values of SA and SAP cell lines to 31 drugs, with three drugs highlighted. **g**, **h** Cell viability in vitro measured by MTT assay for indicated drugs, with IC50 noted ($n = 5$ for each concentration). **i** Western blot showing the upregulation of phospho-HER (Tyr1248 and Tyr1221) signaling in SAP tumors compared with SA tumors ($n = 3$ for each group). In **c–e**, box plots visualize the five-number summary of a data set (minimum, lower quartile, median, upper quartile and maximum). In **g**, **h**, data represent mean ± SD. $^{\#}P > 0.05$, $^{*}P < 0.05$, $^{**}P < 0.01$, $^{***}P < 0.001$, two-sided Student's $t$ test.

direction in the two genotypes, with the exception of phospho-Akt (S473) which was upregulated in SAP tumors but down-regulated in SA tumors (Fig. 6b). Among the commonly upregulated proteins in SA and SAP tumors, proteins involved in metabolic regulation, cell motility and morphology, cell cycle and stemness were notable (Fig. 6c). Interestingly, p16[INK4a], whose overexpression is often found correlated with high-risk HPV infection[44], was upregulated in both SA and SAP tumors (Fig. 6c

and Supplementary Fig. 6a). The alternative product p19[ARF] (not included in RPPA) from the INK4a locus was also upregulated in both tumors (Supplementary Fig. 6a). Another drastically over-expressed protein, CDH3 (i.e. P-Cadherin), was confirmed by IHC (Supplementary Fig. 6b). Among the commonly under-expressed proteins (Fig. 6d), AR was notable because of similar trend observed in clinical PSCC (Fig. 1a). AR downregulation and exclusion from nucleus in SA and SAP tumors was evident by

IHC analysis (Supplementary Fig. 6c), consistent with our result showing AR signaling is dispensable for sustaining PSCC progression (Fig. 1i-l). Using signals from the phospho-specific and total Akt antibodies, the calculated pS473/total-Akt ratio showed significant increase in SAP but not SA tumors (Fig. 6e). This result was validated by western blot (Fig. 6e) and consistent with *Pten*-loss induced PI3K/Akt pathway activation in SAP tumors.

Overexpressed proteins in SA and SAP tumors provide the opportunity for the identification of potential therapeutic targets. We established 3 SA cell lines (SA1, SA2, SA3) and 2 SAP cell lines (SAP1, SAP2) from the two tumor models respectively. Compared with the tumor issues which contain heterogeneous cell populations, these cell lines showed cleaner pattern of respective gene knockouts (Supplementary Fig. 6d, e). We used SA1 and SAP1 cell lines to perform a 42-compound dose titration assays with small molecules that target the overexpressed proteins and related pathways (Supplementary Table 9). These cells showed medium to high sensitivity to 31 drugs (IC50 < 100 μM for both lines), with SAP1 cells displaying comparable or higher IC50 for most drugs except for mubritinib (Fig. 6f). In the validation assays, we confirmed the high sensitivity of both SA1 and SAP1 cells to staurosporine, a potent protein kinase C inhibitor with anti-tumor activity and low toxicity in a liposomally encapsulated form in preclinical models[45]. The higher sensitivity of SAP1 to mubritinib relative to SA1 (Fig. 6h) is likely due to the on-target effect on the active HER2 signaling in SAP tumors (Fig. 6i), because mubritinib is a selective HER2 tyrosine kinase inhibitor[46]. Together, the SA1 and SAP1 cell lines, in conjunction with the GEM models, provide the in vitro and in vivo models to develop therapeutic strategies for the treatment of lethal PSCC.

**Expression convergence of human and mouse PSCC.** In order to determine if the spontaneous PSCC developed in SA and SAP mice resemble human PSCC at the molecular level, we compared the top 200 up- and down-regulated genes in human penile cancer[47] with the mouse transcriptomic data. For the 172 out of the 200 human genes that have the mouse homologs, 108 (63%) and 89 (52%) showed significant difference in SAP and SA models, respectively (Fig. 7a and Supplementary Table 10), demonstrating significant convergence of gene expression pattern. IPA analysis of the Kuasne dataset identified several highly enriched pathways that were also enriched in SA tumors, in particular immune-related pathways such as Agranulocyte Adhesion and Diapedesis ($P = 7.1 \times 10^{-9}$), Leukocyte Extravasation Signaling ($P = 0.003$) and Eicosanoid Signaling ($P = 0.03$) (Supplementary Table 11). Wnt/β-catenin Signaling was also highly enriched (Supplementary Fig. 7, $P = 0.001$), consistent with the enhanced cytoplasmic and nuclear β-catenin level (Supplementary Fig. 2c). IPA identified potential upstream regulators that explain the transcriptional changes of human penile cancer, which included activated targets by pro-inflammatory factors such as TNF, CSF2, IFNγ, IL1B, and inhibited targets by TGFβ1 (Supplementary Table 12). Interestingly, cytokines (IL1A, IL1B, TNF, CXCL1, CCL20) and MMPs (MMP7, 9, 10, 12, 13), upregulated in both human and mouse PSCC, showed no correlation with HPV status (Fig. 7b). The strongly expressed inflammatory genes in both human and mouse PSCC prompted us to examine the immune cell markers in human penile cancer tissues. Compared with adjacent normal penile compartment, the malignant tissue was infiltrated extensively with T cells (CD3+) including the CTL subset (CD8+), as well as macrophages (CD68

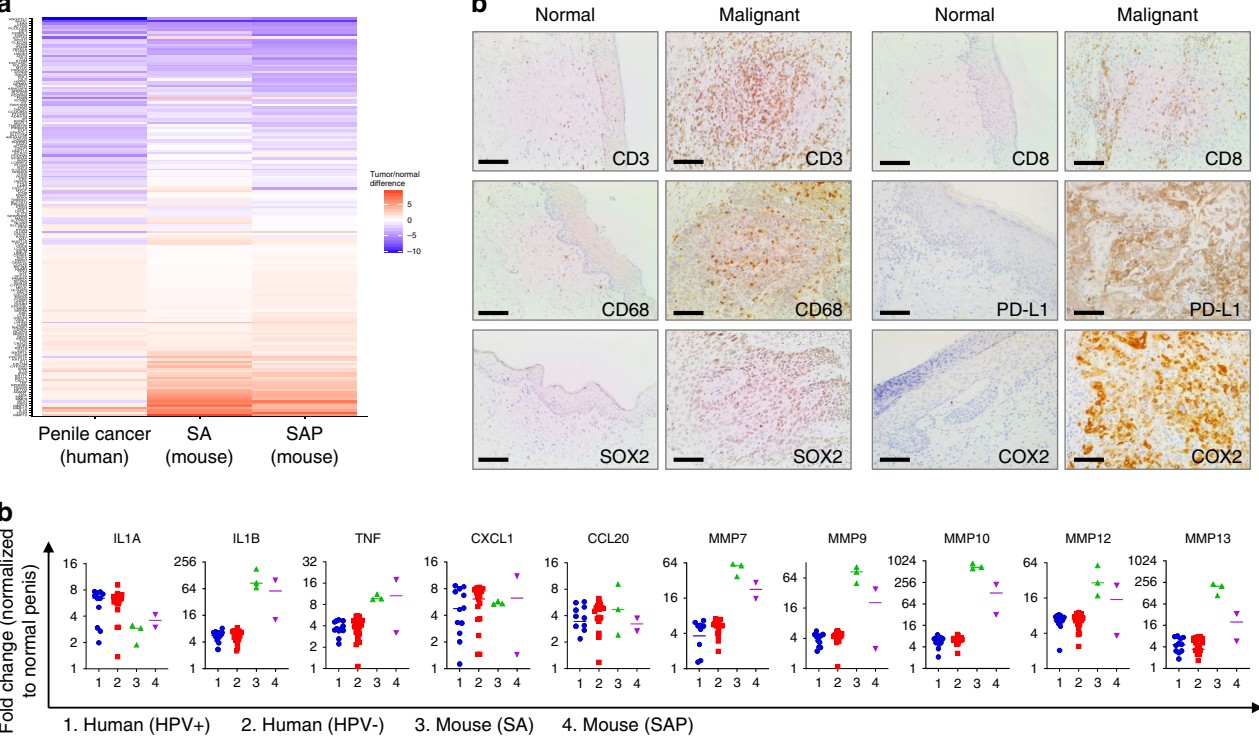

**Fig. 7 Conserved gene expression pattern of human and mouse penile cancers. a** Conserved expression pattern of most up- and down-regulated genes in human penile cancer relative to normal human penis as compared with mouse penile cancer (SA, SAP) relative to normal mouse penis (n = 3, 3 and 2 for WT, SA and SAP, respectively). Human data were from GSE57955[47], n = 39 (12 are HPV+, 25 are HPV−, 2 with unknown HPV status, all referenced to normal glan tissues). **b** Normalized RNA levels of representative upregulated genes in human and mouse PSCC. For human, HPV+ and HPV− cases were plotted separately, but showed no statistically significant differences (n = 12 for human HPV+, 25 for human HPV−, 3 for mouse SA, and 2 for mouse SAP). All data points are shown with the line representing the median. **c** IHC images of selected immune cell markers and signaling molecules in human penile tumors. In all cases, more than half of the eight cases were positive for the marker being stained (n = 8). Scale bar 100 μm.

+) (Fig. 7c). Recent reports showed frequent PD-L1 upregulation in PSCC[48], which was confirmed in our small cohort (Fig. 7c). We also validated that SOX2 and COX2 were upregulated in human PSCC (Fig. 7c). Together, the resemblance of murine PSCC model with human PSCC underscores the clinical relevance of the mouse model.

## Discussion

Our studies build essential resources for translational research in penile cancer. The GEM model of PSCC in our studies is based on the use of PB-Cre4 as the Cre driver in AR-expressed androgen-responsive epithelial cells of the mouse penis. Androgen-AR signaling has a fundamental role in masculinization during development, initiating formation of the prostate, penis and tissues of the male reproductive tract[49]. Transcriptional activity of AR is essential for normal penile growth, as inhibition of androgen in the neonatal phase induces micropenis in mice[50]. Our models indicate that the androgen-sensitive epithelial cells are the potential cells of origin for PSCC, although AR expression is unlikely to be essential for sustaining tumorigenesis based on its diminished level in established PSCC (Fig. 1a). Given the variety of tumor suppressor genes and oncogenes manipulated to model prostate cancer using the PB-Cre4 driver, it is remarkable that concurrent inactivation of Smad4 and Apc were the only reported genetic changes sufficient to drive oncogenesis of mouse penile epithelium. APC loss and WNT pathway activation may also play a key tumorigenic role in human PSCC. Inactivation of APC through mutation, promoter methylation, and loss of heterozygosity is frequent in oral SCC[51–53]. WNT ligand and β-catenin downstream targets (WNT4, MMP7, CCND1, MYC) were found upregulated in PSCC[54]. It is worth noting that activation of WNT/β-catenin targets may be achieved through various mechanisms. For example, loss of function of *FAT1*, caused by recurrent somatic mutations in various cancer types, leads to β-catenin nuclear localization and transcriptional activation of WNT downstream targets[55]. *FAT1* mutation occurs in about 15% of penile cancer[56] and may account for β-catenin activation in penile cancer. In the backdrop of *Apc* deletion, removal of the tumor suppressive function of TGFβ/SMAD (through *Smad4* deletion) and installation of the tumor promoting function of PI3K/AKT/mTOR (through *Pten* deletion) independently promotes the formation of SCC of the mouse penis. In fact, *Smad4* knockout in head and neck epithelia of mice led to development of head and neck SCC[57], which together with our results suggest the general role of *Smad4* as a gatekeeper gene for SCC. We like to note that our model should raise a concern to the many investigators who use PB-Cre4 to model prostate diseases like prostate cancer. Since this Cre driver is active in penile epithelium, attention should be made to whether the prostate phenotype might be influenced by the potentially concurrent phenotype of the penile tissue.

Among the large number of downstream targets of β-catenin activation, we focused on Sox2 and Cox2, both of which are relevant to PSCC. *SOX2* is frequently amplified in SCC, including 32% of penile tumors[58]. PSCC tissues express high level of COX2, PGE2 and EP receptors[28]. SOX2 and COX2 may function as the master regulators to orchestrate intrinsic and extrinsic mechanisms of penile malignancy in the mouse model, in a similar manner as the cooperation of Sox2 and inflammation-inducing Stat3 to transform foregut basal progenitor cells[59]. The essential role of Sox2 in controlling the self-renewal of tumor-initiating cells in SCC have been shown in the skin, head and neck, and lung[27,60,61]. In *Apc*-deficient PSCC, Sox2 may collaborate with β-catenin to regulate the pro-tumor transcriptome, such as upregulation of cyclin D1 as shown previously in breast cancer[62]. We

have shown that Sox2 knockdown attenuated mouse PSCC growth (Fig. 2i–l). Future studies will uncover precise mechanisms of Sox2 in PSCC. The most striking feature of the infiltrating immunoprofile of the SA tumors is the massive CD11b+ Gr1+ MDSCs, which may result from the induction by COX2-mediated PGE2 production in the milieu, a phenomenon observed in both murine and human MDSCs[63–65]. The vulnerability of MDSCs to the inhibition of induction signals (e.g. COX2-prostaglandins) and survival signals (e.g. PI3K) provides opportunities to block these cells and foster a more hospitable environment for immunotherapy to work, as we demonstrated in the SA models (Fig. 4).

The rationale to treat metastatic PSCC with ICB certainly exists, with especially anti-PD-1 or anti-PD-L1 therapies[66]. PD-L1 is detected in 40–60% of PSCC cases and is correlated with poor prognosis and the number of tumor-infiltrating lymphocytes[48]. In a case report, a patient with an HPV-negative PSCC treated with nivolumab experienced a partial response[67]. Currently, there are multiple ICB trials recruiting patients with PSCC (e.g. NCT02837042, NCT02721732, NCT03333616, NCT02834013, and NCT03391479). Our data suggest that better response may be achieved when ICB is combined with targeted therapeutics that eliminates activity from immunosuppressive cells. In a recent Phase I clinical trial on metastatic genitourinary malignancy using cabozantinib with nivolumab and ipilimumab (NCT02496208), manageable toxicity and promising antitumor activity was observed, including partial response for two out four PSCC patients[68]. An important question is whether HPV status should be a criterion for enlisting PSCC patients in ICB trials. Given that PD-L1 expression in PSCC is not correlated with HPV infection[48], together with the observed inflammatory gene signature in both HPV-positive and negative clinical PSCC samples (Fig. 7b), we argue for the concept of treating both HPV-positive and negative PSCC with combined ICB and MDSC-inhibiting drugs.

The PSCC resources developed in our studies include two GEM models and related murine PSCC cell lines. The applications of these reagents could have an impact on developing effective targeted therapy and immunotherapeutic approaches for patients with PSCC in addition to defining pathways of resistance.

## Methods

**Animals.** All animal works performed in this study were approved by The University of Texas MD Anderson Cancer Center Institutional Animal Care and Use Committee and The University of Notre Dame Institutional Animal Care and Use Committee. Acquisition and maintenance of the genetic alleles PB-Cre4, PtenL/L, Smad4L/L, mTmGL/L and ApcL/L were recently described in our studies[31,69]. C57BL/6 and Rag1 knockout were purchased from Jackson Laboratories (stock# 000664 and 003145, respectively). NCr nude mice were purchased from Taconic (stock# NCRNU-M). Since penile cancer is a male disease, all the experimental cohorts used male mice. The mice were maintained in 12 h/12 h light/dark with an ambient temperature of 72 ºF and humidity of 35%. All animals were maintained in pathogen-free conditions and cared for in accordance with the International Association for Assessment and Accreditation of Laboratory Animal Care (AAALAC) policies and certification.

**Cell culture.** Lentivirus packaging cell line 293T (ATCC, CRL-3216) was cultured in DMEM, 10% fetal bovine serum (FBS), 100 U/mL (1x) Penicillin-Streptomycin. Murine penile cancer cell lines SA1, SA2, SA3, SAP1 and SAP2 were established from individual established SA and SAP penile tumors. All the cells were cultured in DMEM, 10% fetal bovine serum (FBS), 100 U/mL (1x) Penicillin-Streptomycin. The cell lines were of the male gender and confirmed free of mycoplasma with MycoAlert Mycoplasma Detection Kit (Lonza).

**Clinical samples.** The eight patient specimen cohort (see Supplementary Table 1) was retrieved from banked archival tissue specimens among patients who had previously consented to have their incidental specimens banked for future research at the University of Texas M.D. Anderson Cancer Center utilizing an approved protocol. Specific blocks were selected based upon the abundance of tissue for additional studies (i.e., usually tumors greater than 1 cm in size) and the patients were further characterized with respect to age, clinical stage, grade, histology, type

of surgery, and Human papillomavirus status. De-identified specimens were provided to the research team under an institutional protocol (PA15-0138) for further characterization under waiver of consent.

**Histology, tissue array, immunostaining, and western blot.** Tissues were fixed in 10% formalin overnight and embedded in paraffin. Hematoxylin & eosin staining was performed using standard protocols on 5 µm paraffin sections. Histologic evaluation of H&E stained sections of mouse penile tumors was performed by two urological pathologists (P.T. & P.R.).

For immunohistochemistry (IHC), 5 µm paraffin sections were deparaffinized, followed by pressure cooker boiling in antigen retrieval solution (10 mM sodium citrate, pH = 6). Subsequent steps followed manual instructions of VECTASTAIN Elite ABC-HRP Kit (Vector Laboratories) with DAB substrate (Vector Laboratories). Primary antibodies are listed in Key Resources Table. IHC slides were scanned with Pannoramic Digital Slide Scanner (3DHISTECH) and images were cropped from virtual slides in Pannoramic Viewer. For tissue array studies of β-catenin IHC of human normal penile tissue and PSCC, we used two commercially available tissue arrays, PE241 and PE2081, from US Biomax, Inc. PE241 had 20 PSCC cases and PE2081 had 174 PSCC cases (3 had no tissues). Together these two arrays had seven adjacent normal tissues. The tissue array slides were deparaffinized and stained with β-catenin antibody (Cell Signaling Technology).

For immunofluorescence (IF) staining, tissues were harvested freshly and cryopreserved in Optimal Cutting Temperature (OCT) Compound. Cryosections were blocked in 5% normal goat serum (Vector Laboratories), and incubated with primary antibodies (see Key Resources Table) overnight at 4 °C. Samples were then incubated with secondary antibodies conjugated with Alexa Fluor 488 or Alexa Fluor 568 (Invitrogen) at RT for 1 h. Slides were mounted with VECTASHIELD mounting medium with DAPI (Vector Laboratories). Stained slides were imaged using Nikon A1R Confocal Laser Microscope and images were managed with ImageJ software.

For western blot, fresh tissues were soaked in ice-cold RIPA buffer (Boston BioProducts) supplemented with protease and phosphatase inhibitors (Roche), and homogenized with Fisherbrand 150 Handheld Homogenizer. Samples were centrifugation at 15,000×g for 10 min at 4 °C. Proteins were resolved by SDS–PAGE, transferred to nitrocellulose membrane and analyzed by immunoblot using primary antibodies listed in the Key Resources table. Anti-rabbit or anti-mouse IgG conjugated to horseradish peroxidase (Cell Signaling Technology) were used as secondary antibodies. Detection was performed with Clarity Max Western ECL Substrate (Bio-Rad) followed by exposure to Amersham ECL Hyperfilm (GE Healthcare).

**Reverse phase protein array.** Reverse phase protein array (RPPA) experiment was performed at the RPPA core at MD Anderson Cancer Center. Total 27 freshly dissected mouse penile normal or tumor samples were probed with 299 antibodies. Briefly, tumor lysates were serially diluted two-fold for 5 dilutions (from undiluted to 1:16 dilution) and arrayed on nitrocellulose-coated slides in an 11 × 11 format. Samples were probed with antibodies by tyramide-based signal amplification approach and visualized by DAB colorimetric reaction. Slides were scanned on a flatbed scanner to produce 16-bit tiff image. Spots from tiff images were identified and the density was quantified by Array-Pro Analyzer. Relative protein levels for each sample were determined by interpolation of each dilution curves from the standard curve (supercurve) of the slide (antibody). These values (given as Log2 values) are defined as Supercurve Log2 (Raw) values. All the data points were normalized for protein loading and transformed to linear value and used for bar graphs. The antibody list, including vendor and catalog information, can be found on the website of the RPPA core at MD Anderson Cancer Center. For hierarchical clustering, normalized signals generated from rabbit and goat antibodies were analyzed using one-way ANOVA (alpha 0.05), and the resulted 117 significant proteins were clustered using average lineage method with Pearson correlation.

**Mass cytometry (CyTOF).** The procedure for antibody staining and gating strategy was described as following[70]. Briefly, single cells from mouse penile tumors were prepared using the Mouse Tumor Dissociation kit (Miltenyl Biotec). All isolated cells were depleted of erythrocytes by hypotonic lysis, and blocked for FcγR using CD16/CD32 antibody (clone 2.4G2, BD Biosciences) and incubated with metal-conjugated antibodies targeting cell surface antigens for 30 minutes at room temperature. Cells were washed and incubated with 5 µM Cell-ID Cisplatin-195Pt (Fluidigm) for 3 min for viability staining. Cells were washed and fixed following the steps of the Foxp3/Transcription Factor Staining Buffer Set (eBioscience). Fixed cells were stained with metal-conjugated antibodies targeting intracellular antigens for 1 hour at room temperature. Cells were washed and incubated with Nucleic Acid Intercalator-Ir (Fluidigm) at 4 °C overnight to stain all the nuclei. The samples were analyzed with CyTOF instrument (Fluidigm) in the Flow Cytometry and Cellular Imaging Core Facility at MD Anderson Cancer Center. CyTOF data were analyzed and visualized using Cytobank (http://www.cytobank.org).

**Therapeutic treatment in SA model.** SA mice develop spontaneous penile tumors. Therefore, the mice were monitored daily for penile prolapse phenotype and, once the tumors reached 2–3 mm in diameter, the mice were randomly assigned to one of the treatment arms of a preclinical trial. Cabozantinib (Selleck Chemicals, S1119) and celecoxib (LC Laboratories, C-1502) were orally administered at daily doses of 30 mg/kg and 200 mg/kg, respectively, daily on a Monday through Friday schedule. Anti-PD1 (clone RMP1-14, BioXcell, BE0146) and anti-CTLA4 (clone 9H10, BioXcell, BE0131) antibodies (or their respective isotype IgG controls) were intraperitoneally administered at 200 µg/injection three times/week. Cisplatin was intraperitoneally injected weekly at 10 mg/kg (equivalent to 40 mg/m$^2$ in human). The duration of drug treatment was 4 weeks. At the endpoint, the penis from each treated mouse was resected and weighed, then fixed in formalin for histological analysis.

**Gene expression profiling and analysis.** Three 5-month-old SA mice were used for harvesting penile tumors. Age-matched WT (wild type) C57BL/6 mice were used for harvesting normal penile epithelium. Total RNA from fresh tissues was extracted using Direct-zol RNA Miniprep Kit (Zymo Research). The quantity and quality of each sample were measured using an Agilent 2100 Bioanalyzer. RNA sequencing was performed at Sequencing and Microarray Facility at MD Anderson Cancer Center using standard stranded RNA-seq protocol on an Illumina HiSeq 4000 (20 million pair-end sequence reads per sample).

The raw reads were aligned to the mouse reference genome build mm10, using Tophat RNASeq alignment software. The aligned reads were verified for quality using FASTQC software. HTseq software was used to summarize the gene expression counts from Tophat alignment data after sorting the BAM files. The raw counts were normalized and differential expression analysis was performed using the DEseq2 package. Cluster analysis was performed with the log transformed normalized count Data. Heatmaps were used to depict the clustering and differential expression analysis results. Significant genes were defined by using a cut-off of 0.01 on the BH corrected P-value and an absolute log2(fold change) value of at least 2. Pathway analysis and upstream regulator analysis were performed with Ingenuity Pathway Analysis (QIAGEN). The raw and normalized data files are deposited by Basepair Tech to Gene Expression Omnibus (GEO) with series entry number GSE130052.

To examine if the differentially expressed genes between mouse penile tumor and normal penile epithelium were also dysregulated in human penile cancer samples, we downloaded published transcriptomic dataset of human penile cancer GSE57955[47] and identified the top 200 up- and down-regulated genes. Among these 200 genes, 172 have mouse homolog genes. Among these 172 genes, 108 (63%) and 89 (52%) genes showed significant difference in SAP and SA models (penile tumor compared with normal mouse penis in the RNA-seq data), respectively. The results were displayed as a clustering heatmap (Fig. 7a). For a number of selected genes, human and mouse gene RNA levels (normalized to normal penile tissues) were plotted to demonstrate consistent cross-species upregulation of these genes (Fig. 7b).

**T cell proliferation assay and suppression by MDSC coculture.** Briefly, to isolate MDSCs from SA penile tumors, tumors were harvest and digested by Mouse Tumor Dissociation kit (Miltenyi Biotec) to obtain the single cells. Total MDSCs were isolated using Mouse Myeloid-Derived Suppressor Cell Isolation Kit (Miltenyi Biotec). Total spleen T cells were isolated from spleen of 2-month-old male C57BL/6 mice (Jackson Laboratory) using the mouse Pan T Cell Isolation Kit II (Miltenyi Biotec). To assess the suppressive activities of MDSCs on T cell proliferation, we first labeled the T cells with 5 µM carboxyfluorescein diacetate succinimidyl ester (CFSE) (Invitrogen). The labeled T cells were stimulated in an antigen-nonspecific manner with anti-CD3 and anti-CD28 antibodies following eBioscience's protocol. MDSCs were added at different ratios to T cells. CFSE intensity was quantified 72 h later with BD LSRFortessa Cell Analyzer. Viable CD3$^+$ T cells with CFSE peaks located left of the highest peak (i.e. no proliferation, thus no decline of CFSE intensity) were counted as CFSE$^{low}$ (i.e. proliferative) cells.

**Sox2 knockdown and in vivo experiments.** To knockdown Sox2 with shRNA, scramble control SHC202 or Sox2-specific shRNA (Sigma-Aldrich, TRCN0000416106 as #2, TRCN0000420955 as #3) lentiviruses were packaged with 293T cells and used to infect SA1 cell line to generate puromycin-resistant stable sublines. The knockdown was confirmed by western blot. For in vivo tumorigenesis, $1 \times 10^6$ cells in 100 µl PBS were injected subcutaneously in the dorsal skin of male 6-week-old NCr nude mice. Tumor dimensions were measured with a digital caliper and tumor volume was determined as 1/2 (length × width$^2$). The mice were euthanized at the experimental endpoint, and tumors were dissected, weighted and photographed.

**Inhibitor screening using mouse penile cancer cell lines.** Inhibitors were prepared as 10 mM stock in DMSO or water according to manufacturer's instructions. SA1 and SAP1 cell lines were seeded at $10^4$ cells/well in 96-well plates in the absence or presence of inhibitors at 0.2, 2.0, and 20 µM in triplicates. In all, 48 h later, 3-(4,5-Dimethyl-2-thiazolyl)-2,5-diphenyl-2H-tetrazolium bromide (MTT) was added to the wells at final concentration of 0.5 mg/ml. The plates were gently shaken at 500 rpm for 1 min, and incubated at 37 °C for 4 h. The medium was

removed completely, and 100 µl DMSO was added into each well. The plates were gently shaken at 1000 rpm for 5 min to mix well the colored solution, and placed to microplate reader (Epoch 2, BioTek) to read 540 nm and 690 nm (reference). A confirmative second round of MTT assay was performed with a more refined range of inhibitor concentrations to determine IC50 more precisely.

**Penile orthotopic injection**. Male Rag1 knockout males are anesthetized by inhalation of 2% isoflurane, then placed on the back with arms and legs extended. The urogenital region is prepared for surgery with alcohol swab. Sterile insulin syringe with 30-gauge needle is loaded with 50 µl of $1 \times 10^6$ mouse penile cancer cells. With one hand keeping the glans penis stretched, from the ventral side of the glans penis, the other hand aims the needle to the superficial epithelial spine layer and keeps the bevel parallel with the glans axis so that the needle does not penetrate the glans. Cells are slowly injected with the formation of a small bump at the injection site as the indication of successful injection.

**Radical prostatectomy**. Anesthetized mouse is placed on its back with arms and legs extended. An operating stereomicroscope is used for the entire surgery. Access to the pelvis is via a median abdominal incision (2.5 cm). The prostate ventral lobes and dorsolateral lobes are bluntly separated from the urethra using a cotton swab. The anterior prostate is dissected using the seminal vesicles as landmarks, and the prostate is then detached from the dorsal urethra.

**Surgical castration**. The mouse was anesthetized and both testes of the mouse were pushed down into the scrotal sacs, followed by ~1 cm incision through the skin along the midline of the scrotal sac. Then a 5 mm incision was made in the membrane on one side of the midline. The testis was pushed out, dissected away from the fat pad, and removed with scissors. After that, the fat pad was pushed back into the scrotal sac. Repeat above steps for the other testis. Buprenorphine was used for analgesia after all surgical procedures. If it is sham-operation, there is no removal of the testis.

**Prostate orthotopic injection**. The mouse was anesthetized and placed on its back with arms and legs extended. The lower abdomen wall of the mouse was wiped with 70% ethanol followed by betadine swab. The prostate anterior lobe was exposed with a low midline abdominal incision about 1 cm, and 30-gauge needle was used to inject of $5 \times 10^5$ cells in 50 µl PBS. The prostate was returned to the peritoneum. Buprenorphine was used for analgesia after all surgical procedures.

**Statistics and reproducibility**. Data were presented as mean ± standard deviation (SD) unless indicated otherwise. Number of biological samples (n) for the experiments was denoted in figure legends. Results showing representative experiments (such as micrographs) were repeated independently at least three times with similar results. Unless indicated otherwise, two-sided Student's $t$ test assuming two-tailed distributions was used to calculate statistical significance between groups (no assumption was made that variance was similar between the groups that are being statistically compared). $P < 0.05$ was considered statistically significant. Data analysis was assisted with Graphpad Prism version 7, Cytobank 6.3.1, R version 3.3.3, and Ingenuity Pathway Analysis (QIAGEN).

**Reagents and resources**. All the reagents and resources used and generated in this study are listed in Supplementary Table 13.

**Reporting summary**. Further information on research design is available in the Nature Research Reporting Summary linked to this article.

## Data availability

The RNAseq data generated in the study have been deposited and available for public access in the Gene Expression Omnibus (GEO) database under the accession code GSE130052. The GSE57955 data referenced during the study are available in a public repository from the GEO website. All other data supporting the findings of this study are available within the article and its supplementary information files and from the corresponding author upon reasonable request. A reporting summary for this article is available as a Supplementary Information file.

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

## Acknowledgements

We thank the members of our laboratories for helpful discussions. The research reported in this publication was supported by Award Grant Number# CA096297/CA096300 from the National Cancer Institute of the National Institutes of Health. The content is solely the responsibility of the authors and does not necessarily represent the official views of the National Institutes of Health. X.L. was partially supported from Grant Numbers KL2 TR002530 and UL1 TR002529 (A. Shekhar, PI) from the National Institutes of Health, National Center for Advancing Translational Sciences, Clinical and Translational Sciences Award, Susan G. Komen Grant # CCR18548293, Boler Family Foundation endowment at University of Notre Dame. T.H. was partly supported by a graduate fellowship from China Scholarship Council. J.C. is supported by an ASCO Conquer Cancer Foundation Young Investigator Award.

## Author contributions

Conceptualization, X.L. and C.A.P.; methodology, X.L. and C.A.P.; investigation, T.H., X. C., J.C., A.S., X.S., P.D., Y.L., S.F., and X.L.; formal analysis, P.T., P.R., M.G., G.M., L. Z., Y. X., L. H. and Xuemin L.; resources, P.T., P.R. and C.A.P.; writing—original draft, X.L.; writing—review and editing, X.L. and C.A.P.; funding acquisition, Y.A.W., M.M.F., C.A. P. and X.L. Supervision, Y.A.W., M.M.F., C.A.P. R.A.D., and X.L.

## Competing interests

The authors declare no competing interests.
