## [Peer Review File · Nature Communications]

REVIEWERS' COMMENTS:

Reviewer #1 (Remarks to the Author):

This is a well designed and elegantly conducted study. It is well written. The conclusions are supported by the results. The topic is of interest and timely. Although the incidence of PSCC in the US is low, Penile cancer is a major healthcare concern in Latin America and other under developed regions. Further understanding of molecular pathways of PSCC oncogenesis and identification of therapeutic targets are of interest.

Reviewer #4 (Remarks to the Author):

This revised manuscript by Huang et al. adequately addresses the major concern of the previous version that the phenotype might be confounded by a prostate phenotype.

Removing the prostate and doing the androgen deprivation are nice additions.

It is of interest (and should be of concern to the many investigators that use this allele...) that a Cre allele that is widely used for analyses of prostate phenotypes can lead to penil cancer.

It be beneficial to the community if the authors could comment about this in the discussion (and perhaps in the text). This point is really downplayed.

Reviewer#4 (Comments on authors responses to reviewer#2):

I just read this and I think the comments of Reviewer 2 are very well addressed. I have no concerns.

Responses to Reviewer' Comments

Reviewer #1 (Remarks to the Author):

This is a well designed and elegantly conducted study. It is well written. The conclusions are supported by the results. The topic is of interest and timely. Although the incidence of PSCC in the US is low, Penile cancer is a major healthcare concern in Latin America and other under developed regions. Further understanding of molecular pathways of PSCC oncogenesis and identification of therapeutic targets are of interest.

Answer: We appreciate the positive comments from Reviewer #1.

Reviewer #4 (Remarks to the Author):

This revised manuscript by Huang et al. adequately addresses the major concern of the previous version that the phenotype might be confounded by a prostate phenotype. Removing the prostate and doing the androgen deprivation are nice additions. It is of interest (and should be of concern to the many investigators that use this allele...) that a Cre allele that is widely used for analyses of prostate phenotypes can lead to penil cancer.

It be beneficial to the community if the authors could comment about this in the discussion (and perhaps in the text). This point is really downplayed.

Answer: We are glad that the reviewer recognized the significance of the added androgen deprivation and castration experiments. Regarding the suggestion on adding a comment about the PB-Cre4 driver, we added the following sentences at the end of the first paragraph in Discussion (Page 16):

“We like to note that our model should raise a concern to the many investigators who use PB-Cre4 to model prostate diseases like prostate cancer. Since this Cre driver is active in penile epithelium, attention should be made to whether the prostate phenotype might be influenced by the potentially concurrent phenotype of the penile tissue.”

Reviewer#4 (Comments on authors responses to reviewer#2):

I just read this and I think the comments of Reviewer 2 are very well addressed. I have no concerns.

Answer: We are glad that the reviewer's comments were addressed satisfactorily.